

# Genetic analyses reveal population structure and recent decline in leopards (*Panthera pardus fusca*) across the Indian subcontinent

Supriya Bhatt[1,*], Suvankar Biswas[1,*], Krithi Karanth[2,3], Bivash Pandav[4] and Samrat Mondol[1]

[1] Animal Ecology and Conservation Biology, Wildlife Institute of India, Dehradun, India
[2] Centre for Wildlife Studies, Bengaluru, India
[3] Nicholas School of Environment, Duke University, Durham, United States of America
[4] Endangered Species Management, Wildlife Institute of India, Dehradun, India
[*] These authors contributed equally to this work.

## ABSTRACT

**Background**. Large carnivores maintain the stability and functioning of ecosystems. Currently, many carnivore species face declining population sizes due to natural and anthropogenic pressures. The leopard, *Panthera pardus*, is probably the most widely distributed and highly adaptable large felid globally, still persisting in most of its historic range. However, we lack subspecies-level data on country or regional scale on population trends, as ecological monitoring approaches are difficult to apply on such wide-ranging species. We used genetic data from leopards sampled across the Indian subcontinent to investigate population structure and patterns of demographic decline.

**Methods**. We collected faecal samples from the Terai-Arc landscape of northern India and identified 56 unique individuals using a panel of 13 microsatellite markers. We merged this data with already available 143 leopard individuals and assessed genetic structure at country scale. Subsequently, we investigated the demographic history of each identified subpopulations and compared genetic decline analyses with countrywide local extinction probabilities.

**Results**. Our genetic analyses revealed four distinct subpopulations corresponding to Western Ghats, Deccan Plateau-Semi Arid, Shivalik and Terai region of the north Indian landscape, each with high genetic variation. Coalescent simulations with microsatellite loci revealed a possibly human-induced 75–90% population decline between ∼120–200 years ago across India. Population-specific estimates of genetic decline are in concordance with ecological estimates of local extinction probabilities in these subpopulations obtained from occupancy modeling of the historic and current distribution of leopards in India.

**Conclusions**. Our results confirm the population decline of a widely distributed, adaptable large carnivore. We re-iterate the relevance of indirect genetic methods for such species in conjunction with occupancy assessment and recommend that detailed, landscape-level ecological studies on leopard populations are critical to future conservation efforts. Our approaches and inference are relevant to other widely distributed, seemingly unaffected carnivores such as the leopard.

Corresponding author
Samrat Mondol, samrat@wii.gov.in

# INTRODUCTION

Large carnivores are critical to ecosystem structure and functioning and their absence can lead to significant changes in them and may affect trophic dynamics, resulting in cascading effects (*Estes et al., 2011*; *Ripple et al., 2014*). Growing natural and anthropogenic pressures in the form of climate change, habitat loss and prey depletion, wildlife trade and human-wildlife conflicts are pushing large carnivores into ever-shrinking habitat islands and severely exacerbating their endangered status, and in some cases extinction (*Schipper et al., 2008*; *Karanth et al., 2010*). Recent assessments of the conservation status indicate alarming rates of population decline for many carnivores at a global scale. Specifically, the families *Felidae*, *Canidae* and *Ursidae* are under severe threat across the globe (*Wolf & Ripple, 2017*).

The leopard (*Panthera pardus*) represents the most widely distributed and adaptable member of the family *Felidae*. The historical range of leopards spanned across nearly 35,000,000 km$^2$ area covering all of sub-Saharan and north Africa, the Middle East and Asia Minor, South and Southeast Asia, and the Russian Far East (*Uphyrkina et al., 2001*; *Jacobson et al., 2016*). However, their current distribution and numbers have significantly decreased across the range due to habitat loss, prey depletion, conflict and poaching over the last century (*Jacobson et al., 2016*). Recent meta-analyses of leopard status and distribution suggest 48–67% range loss for the species in Africa and 83–87% in Asia (*Jacobson et al., 2016*), making them among the top ten large carnivore species most-affected by range contraction (*Wolf & Ripple, 2017*). This has resulted in changing the species status from 'Near Threatened' to 'Vulnerable' by IUCN (*Stein et al., 2016*). Despite continuously decreasing numbers and range, their ubiquitous presence across human habitations leads to misconceptions regarding their current abundance.

Among all the subspecies, the Indian leopard (*P. p. fusca*) retains the largest population size and range outside Africa (*Jacobson et al., 2016*). In the Indian subcontinent poaching and conflict are major threats to leopard populations (*Athreya et al., 2010*; *Raza et al., 2012*). Leopards also frequently occur outside protected areas, increasing their vulnerability to conflict with humans (*Athreya et al., 2010*; *Naha, Sathyakumar & Rawat, 2018*). Unfortunately, there is still a paucity of information on their population and demography at regional and global scales. Few earlier studies have assessed the subspecies status (*Asad et al., 2019*; *Farhadinia et al., 2015*; *Paijmans et al., 2018*) and genetic diversity (*Uphyrkina et al., 2001*; *Dutta et al., 2013*; *Mondol et al., 2014*) of leopards in the Asian region including India, but comprehensive data is lacking. Much of our knowledge on leopard ecology and demography in the Indian subcontinent come from location-specific studies (*Karanth & Sunquist, 2000*; *Chauhan et al., 2005*; *Harihar, Pandav & Goyal, 2009*; *Wang & Macdonald, 2009*; *Kalle et al., 2011*; *Grant, 2012*; *Mondal et al., 2012*; *Dutta et al., 2012*; *Dutta et al., 2013*; *Thapa et al., 2014*; *Borah et al., 2014*; *Selvan et al., 2014*;

*Pawar et al., 2019*). In India, the latest estimate of leopards in the forested habitats of 14 tiger-inhabiting states is 7910 (SE 6566-9181) (*Jhala, Qureshi & Gopal, 2015*). As leopards do survive in highly human populated and modified areas (*Athreya et al., 2010*) this estimate is likely to be minimal and incomplete. Further, recent studies in the Indian subcontinent provide contradictory patterns of local population trends. For example, historical records and occupancy estimation models based on ecological data and field observations by *Karanth et al. (2010)* estimated high local extinction probabilities of leopards across the subcontinent, and *Athreya et al. (2010)* reported higher rates of recent conflict incidences and related mortality at local scales. Other ecological (*Harihar, Pandav & Goyal, 2011*) as well as population genetic studies of demographic history (*Dutta et al., 2013*) suggest stable or increased leopard populations at a landscape scale. However, lack of detailed, systematic field data makes it difficult to generate accurate population estimates as well as demographic patterns at landscape scales.

In this paper, we used faecal samples to assess leopard genetic variation, population structure and demographic history in the Indian subcontinent. More specifically, we investigated (1) extent of genetic variation in leopard that persists across the Indian subcontinent; (2) population structure of leopards at country scale; (3) the demographic history of leopards by assessing recent changes in population size and finally (4) compared the finding of genetic decline analyses with countrywide local extinction probabilities. We interpreted our results in the context of local extinction probabilities as estimated in *Karanth et al. (2010)*. We addressed these questions using genetic data generated using 13 polymorphic microsatellite loci from leopard faecal samples collected across different landscapes of India.

## METHODS

### Research permissions and ethical considerations

All required permissions for our field surveys and biological sampling were provided by the Forest Departments of Uttarakhand (Permit no: 90/5-6), Uttar Pradesh (Permit no: 1127/23-2-12(G) and 1891/23-2-12) and Bihar (Permit no: Wildlife-589). Due to non-invasive nature of sampling, no ethical clearance was required for this study.

### Sampling

To detect population structure and past population demography it is important to obtain genetic samples from different leopard habitats all across the study area. In this study, we used leopard genetic data generated from non-invasive samples collected across the Indian subcontinent. We conducted extensive field surveys across the Indian part of Terai-Arc landscape (TAL) covering the north-Indian states of Uttarakhand, Uttar Pradesh and Bihar between 2016–2018. This region has already been studied for large carnivore occupancy using traditional camera trapping as well as field surveys (*Johnsingh et al., 2004*; *Harihar, Pandav & Goyal, 2009*; *Jhala, Qureshi & Gopal, 2015*; *Chanchani et al., 2016*). We foot surveyed all existing trails covering the entire region to collect faecal samples. Number of trails walked in a particular area was decided based on existing knowledge of leopard presence by the local people and frontline staff members of the sampling team. We

collected a total of 778 fresh large carnivore faecal samples. These samples were collected from both inside ($n = 469$) and outside ($n = 309$) protected areas from different parts of this landscape. In the field, the samples were judged as large carnivores based on several physical characteristics such as scrape marks, tracks, faecal diameter etc. All faecal samples were collected in wax paper and stored individually in sterile zip-lock bags and stored inside dry, dark boxes in the field for a maximum of two weeks period (*Biswas et al., 2019*). All samples were collected with GPS locations and were transferred to the laboratory and stored in $-20\,°C$ freezers until further processing.

In addition to the north Indian samples collected in this study, we used genetic data previously described in *Mondol et al. (2015)*, representing mostly the Western Ghats and central Indian landscape. The data was earlier used in forensic analyses to assign seized leopard samples to their potential geographic origins in India (*Mondol et al., 2015*). Out of the 173 individual leopards described in the earlier study, we removed data from related individuals and samples with insufficient data ($n = 30$) and used the remaining 143 samples for analyses in this study. These samples were collected from the states of Kerala ($n = 5$), Tamil Nadu ($n = 4$), Karnataka ($n = 53$), Andhra Pradesh ($n = 3$), Madhya Pradesh ($n = 12$), Maharashtra ($n = 46$), Gujarat ($n = 2$), Rajasthan ($n = 5$), Himachal Pradesh ($n = 8$), Jharkhand ($n = 1$), West Bengal ($n = 2$) and Assam ($n = 2$), respectively. The sample locations are presented in Fig. 1.

## DNA extraction, species and individual identification

For all field-collected faecal samples, DNA extraction was performed using protocols described in *Biswas et al. (2019)*. In brief, each frozen faeces was thawed to room temperature and the upper layer was swabbed twice with Phosphate buffer saline (PBS) saturated sterile cotton applicators (HiMedia). The swabs were lysed with $30\,\mu l$ of Proteinase K (20 mg/ml) and $300\,\mu l$ of ATL buffer (Qiagen Inc., Hilden, Germany) overnight at $56\,°C$, followed by Qiagen DNeasy tissue DNA kit extraction protocol. DNA was eluted twice in $100\,\mu l$ preheated 1X TE buffer. For every set of samples, extraction negatives were included to monitor possible contaminations.

Species identification was performed using leopard-specific multiplex PCR assay with NADH4 and NADH2 region primers described in *Mondol et al. (2014)* and cytochrome b primers used in *Maroju et al. (2016)*. PCR reactions were done in $10\,\mu l$ volumes containing $3.5\,\mu l$ multiplex buffer mix (Qiagen Inc., Hilden, Germany), $4\,\mu M$ BSA, $0.2\,\mu M$ primer mix and $3\,\mu l$ of scat DNA with conditions including initial denaturation ($95\,°C$ for 15 min); 40 cycles of denaturation ($94\,°C$ for 30 s), annealing ($T_a$ for 30 s) and extension ($72\,°C$ for 35 s); followed by a final extension ($72\,°C$ for 10 min). Negative controls were included to monitor possible contamination. Leopard faeces were identified by viewing species-specific bands of 130 bp (NADH4) and 190 bp (NADH2) (*Mondol et al., 2014*) and 277 bp (cytochrome b) (*Maroju et al., 2016*) in 2% agarose gel.

For individual identification, we used the same panel of 13 microsatellite loci previously used in *Mondol et al. (2014)* (Table 1). To generate comparable data with the samples used from earlier study by *Mondol et al. (2014)* we employed stringent laboratory protocols. All PCR amplifications were performed in $10\,\mu l$ volumes containing $5\,\mu l$ Qiagen multiplex PCR

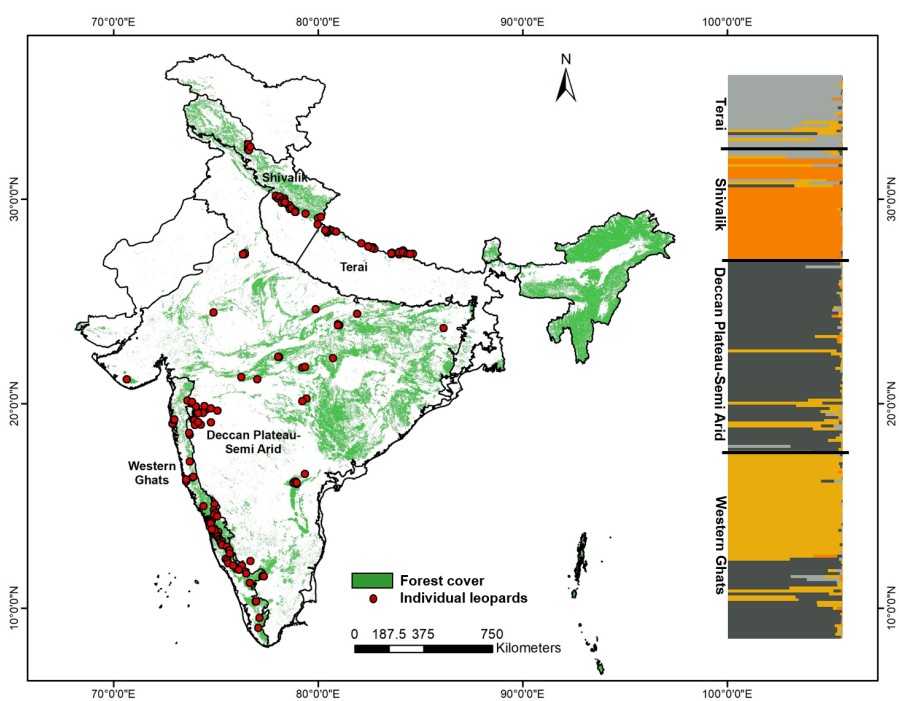

**Figure 1 Genetic sampling and leopard population structure across the Indian subcontinent with forest cover map and leopard sampling locations used in this study.** The map also shows the inferred biogeographic leopard habitats based on genetic structure, as found in this study and corroborative leopard genetic clusters indicated by program STRUCTURE (based on 13 microsatellite loci).

buffer mix (QIAGEN Inc., Hilden, Germany), 0.2 μM labelled forward primer (Applied Biosystems, Foster City, CA, USA), 0.2 μM unlabelled reverse primer, 4 μM BSA and 3 μl of the faecal DNA extract. The reactions were performed in an ABI thermocycler with conditions including initial denaturation (94 °C for 15 min); 45 cycles of denaturation (94 °C for 30 s), annealing ($T_a$ for 30 s) and extension (72 °C for 30 s); followed by final extension (72 °C for 30 min). Multiple primers were multiplexed to reduce cost and save DNA (Table 1). PCR negatives were incorporated in all reaction setups to monitor possible contamination. The PCR products were analyzed using an automated ABI 3500XL Bioanalyzer with LIZ 500 size standard (Applied Biosystems, Foster City, CA, USA) and alleles were scored with GENEMAPPER version 4.0 (Softgenetics Inc., State Collage, PA, USA). During data generation from field-collected samples we used one reference sample (genotyped for all loci) from the earlier study for genotyping. As the entire new data is generated along with the reference sample and the alleles were scored along with the reference genotypes, the new data (allele scores) were comparable with earlier data for analyses.

To ensure high quality multi-locus genotypes from faecal samples, we followed a modified multiple-tube approach in combination with quality index analyses (*Miquel et al., 2006*) as described previously for leopards by *Mondol et al. (2009a)* and *Mondol et al. (2014)*. All faecal samples were amplified and genotyped four independent times for all the

**Table 1  Genetic diversity and genotyping error details for the leopard samples collected across Terai-Arc landscape ($n = 56$) in this study.** A total of 13 microsatellite loci were used. Data from these samples have been added to earlier leopard forensic data described in *Mondol et al. (2014)*.

| Locus | Repeat length | $N_A$ | Allelic size range | $H_E$ | $H_O$ | Null allele | Allelic dropout | False allele | HWE | Reference |
|---|---|---|---|---|---|---|---|---|---|---|
| FCA230 | 2 | 16 | 44 | 0.87 | 0.69 | 0.18 | 0.001 | 0.005 | Yes | *Menotti-Raymond et al. (1999)* |
| FCA309 | 2 | 17 | 42 | 0.85 | 0.70 | 0.22 | 0.004 | 0.004 | Yes | *Menotti-Raymond et al. (1999)* |
| FCA232 | 2 | 15 | 36 | 0.83 | 0.68 | 0.19 | 0.007 | 0.013 | Yes | *Menotti-Raymond et al. (1999)* |
| FCA090 | 2 | 16 | 34 | 0.87 | 0.66 | 0.30 | 0.007 | 0.002 | Yes | *Menotti-Raymond et al. (1999)* |
| FCA052 | 2 | 14 | 32 | 0.85 | 0.77 | 0.19 | 0.004 | 0.006 | Yes | *Menotti-Raymond et al. (1999)* |
| FCA672 | 2 | 20 | 40 | 0.87 | 0.74 | 0.05 | 0.0 | 0.001 | Yes | *Menotti-Raymond et al. (1999)* |
| FCA279 | 2 | 16 | 30 | 0.88 | 0.76 | 0.08 | 0.001 | 0.003 | Yes | *Menotti-Raymond et al. (1999)* |
| FCA126 | 2 | 16 | 32 | 0.89 | 0.70 | 0.36 | 0.004 | 0.001 | Yes | *Menotti-Raymond et al. (1999)* |
| msFCA391 | 4 | 10 | 36 | 0.86 | 0.64 | 0.19 | 0.009 | 0.007 | Yes | *Mondol et al. (2012)* |
| msHDZ170 | 2 | 13 | 42 | 0.83 | 0.53 | 0.30 | 0.002 | 0.002 | Yes | *Mondol et al. (2012)* |
| msFCA441 | 4 | 12 | 52 | 0.82 | 0.52 | 0.23 | 0.006 | 0.003 | Yes | *Mondol et al. (2012)* |
| msFCA506 | 2 | 19 | 56 | 0.89 | 0.69 | 0.25 | 0.008 | 0 | Yes | *Mondol et al. (2012)* |
| msFCA453 | 4 | 7 | 32 | 0.68 | 0.61 | 0.25 | 0.006 | 0.007 | Yes | *Mondol et al. (2012)* |
| Mean (SD) | | 14.69 (3.41) | 39.08 (7.71) | 0.84 (0.05) | 0.67 (0.07) | 0.21 | 0.005 | 0.004 | | |

**Notes.**

$N_A$, No. of alleles; $H_E$, Expected heterozygosity; $H_O$, Observed heterozygosity; HWE, Hardy-Weinberg Equilibrium.

loci. Samples producing identical genotypes for at least three independent amplifications (or a quality index of 0.75 or more) for each loci were considered reliable and used for all further analysis, while the rest were discarded.

## Analysis

For each locus, we calculated average amplification success as the percent positive PCR (*Broquet & Petit, 2004*) after four repeats across all samples. We quantified allelic dropout and false allele rates manually as the number of dropouts or false alleles over the total number of amplifications, respectively (*Broquet & Petit, 2004*), as well as using MICROCHECKER v 2.2.3 (*Van Oosterhout et al., 2004*). The false allele frequency is calculated for both homozygous and heterozygous genotypes as the ratio of the number of amplifications having one or more false alleles at a particular locus and the total number of amplifications while allele dropout rate (ADO) is calculated as the ratio between the observed number of amplifications having loss of one allele and the number of positive amplifications of heterozygous individuals.

Post data quality assessment we selected only those samples with good quality data for at least nine or more loci (out of 13) for further analyses. We used the identity analysis module implemented in program CERVUS (*Kalinowski, Taper & Marshall, 2007*) to identify identical genotypes (or recaptures) by comparing data from all samples. All genetic recaptures were removed from the data set. GIMLET (*Valiere, 2002*) was used to calculate the $PID_{(sibs)}$ for all the unique individuals. Following this, any allele having less than 10% frequency across all amplified samples were rechecked for allele confirmation. ARLEQUIN (*Excoffier, Laval & Schneider, 2005*) was used to determine Hardy Weinberg

equilibrium and linkage disequilibrium for all the loci. Finally, to avoid the effects of related individuals in all analyses, we used the program GENECLASS 2.0 (*Piry et al., 2004*) to select out related individuals in our samples.

To determine the genetic structure of leopards across the Indian subcontinent we used a Bayesian clustering approach implemented in program STRUCTURE (*Pritchard, Stephes & Donnelly, 2000*; *Falush, Stephens & Pritchard, 2003*). We performed 10 independent analyses for each K values between one and ten, using 450,000 iterations and a burn-in of 50,000 assuming correlated allele frequencies. The optimal value of K was determined using STRUCTURE HARVESTER web version (*Earl & VonHoldt, 2012*). Further, we used multivariate analyses approach implemented in program Discriminant Analysis of Principal Component (DAPC) (*Jombart, Devillard & Balloux, 2010*) to identify genetic clusters in our data. This approach transforms the genetic data into principal components, followed by clustering to define group of individuals with a consideration of minimum within group variation and maximum between group variations among the clusters. The analyses were conducted using adegenet package 2.1.1 in R studio 1.1.453 (*R Development Core Team, 2014*) where optimal number of clusters were determined through the Bayesian Information Criterion (*Jombart, Devillard & Balloux, 2010*). Subsequent summary statistics were calculated in ARLEQUIN 3.1 (*Excoffier, Laval & Schneider, 2005*) and indices of overall genetic differentiation (pairwise $F_{st}$) were estimated using GenAlEx version 6.5 (*Peakall & Smouse, 2012*), dividing the leopard populations according to the STRUCTURE results across the Indian subcontinent. The divisions were based on Q-values (estimated proportions of ancestry) calculated in STRUCTURE, where we used $Q > 0.75$ as threshold for assigning individuals to a particular population (*Mora et al., 2010*). Additionally, compression of expected heterozygosity (or $G_{st}$) (*Nei, 1973*) between four leopard sub-populations was calculated in GenAlEx version 6.5 (*Peakall & Smouse, 2012*). Finally, HP-RARE 1.0 (*Kalinowski, 2005*) was used to estimate private alleles within each subpopulation.

## Demography analyses

Demographic analyses were performed with different genetic subpopulations of leopards based on the results from STRUCTURE analyses. We used a number of different approaches to detect past population demography for leopards. The first two qualitative approaches use summary statistics to detect population size changes, whereas the quantitative approach is a likelihood-based Bayesian algorithm. The summary statistic-based methods used were the Ewens, Watterson, Cornuet and Luikart method implemented in program BOTTLENECK (*Cornuet & Luikart, 1996*), and the Garza-Williamson index or M ratio (*Garza & Williamson, 2001*) implemented in program ARLEQUIN 3.1. The quantitative Bayesian approach used was implemented in the program MSVAR 1.3 (*Storz & Beaumont, 2002*).

### (a) The Ewens, Watterson, Cornuet and Luikart (EWCL) approach:

This approach allows the detection of population size changes using two summary statistics of the allele frequency spectrum, number of alleles ($N_A$) and expected heterozygosity ($H_e$) across different mutational models. Simulations are performed to obtain the expected

distribution of $H_e$ for a demographically stable population under three mutation models: infinite allele model (IAM), single stepwise model (SMM) and two-phase model (TPM) and the values are then compared to the real data values. This method can detect departures from mutation-drift equilibrium and neutrality, which can be explained by any departure from the null model, including selection, population growth or decline. More importantly, consistent results from independent loci could be attributed to demographic events over selection. For simulations with TPM model, we used two different (5% and 30%) multi-step mutation events for leopards.

### (b) The Garza-Williamson index/M ratio approach:

This approach allows the detection of population decline using two summary statistics of the allele frequency spectrum, number of alleles ($N_A$) and the allelic size range. The basic principle behind this approach is in a reducing population, the expectation of the reduction of number of alleles is much higher than the reduction of allelic size range. Thus, the ratio between the number of alleles and the allelic size range is expected to be smaller in recently reduced populations than in equilibrium populations.

### (c) The Storz and Beaumont approach:

This approach is an extension of Beaumont's approach (*Beaumont, 1999*) that assumes a stable population of size $N_1$ started to change (either decrease or increase) $T_a$ generations ago to the current population size $N_0$. This change in the population size is assumed to be at an exponential scale under stepwise mutation model (SMM), at a rate $y = 2N_0m$, where m is the mutation rate per locus per generation. This Bayesian approach uses the information from the full allelic distribution in a coalescent framework to estimate the posterior probability distribution, allowing quantification of effective population sizes $N_0$ and $N_1$, rather than their ratio (as in *Beaumont, 1999*) along with T, time since the population change. In this approach, prior distributions for $N_0$, $N_1$, T and $\mu$(mutation rate) are assumed to be log normal. The mean and the standard deviations of these prior log normal distributions are drawn from prior (or hyperpriors) distributions. A Markov Chain Monte Carlo (MCMC) algorithm is used to generate samples from the posterior distribution of these parameters. We used wide uninformative priors to perform multiple runs for this approach (Table S1). For minimal effect towards the posterior distributions variances for the prior distributions were kept large. A total number of 2 million iterations were performed for each run.

The generation time for leopards are known to be about 4–5 years (*Dutta et al., 2013*) and we used a five-year generation time for all analyses.

## Estimation of leopard extinction probability

To understand extinction probability across various biogeographic zones of India we analysed patterns and determinants of leopard occurrence as described in *Karanth et al. (2009)* and *Karanth et al. (2010)*. In this study, we have just divided the earlier information available for leopards for different genetic subpopulations. We applied a grid-based approach to determine current distribution patterns for leopards, where the selection of grids was based on prior information of leopard presence. This involved

collating presence-absence information from more than 100 Indian wildlife experts along with historical information of leopard presence involving hunting locations and other taxidermy and museum records. Each grid cell was an average of 2,818 km$^2$ in size and we used data from 1,229 grid cells covering 3,463,322 km$^2$ area of the Indian subcontinent. This study applied occupancy modelling to examine the influence of ecological and social covariates on patterns of leopard occupancy. We used a maximum likelihood approach for leopard occupancy in PRESENCE. V.2.0 program (*Hines, 2006*). Covariates likely to influence leopard distribution modelled included presence and extent of protected areas, land cover-land use characteristics, human cultural tolerance and population density. Data for protected areas was retrieved from the World Database on protected areas (http://www.unep-wcmc.org) and topographic maps. Land cover- land use data were derived from Global Land Cover Facility (2000) and further refined based on *Roy et al. (2006)* and *Joshi et al. (2006)*. A human tolerance index that characterized different Indian states from most to least tolerant was developed based on knowledge about society-culture, law enforcement, hunting patterns and prior field experiences (for details see *Karanth et al., 2009*; *Karanth et al., 2010*). Human population density data were derived from LandScan Global Population Data 2000 (http://www.ornl.gov/gist). Based on existing information on species' ecology we predicted higher occupancy in protected areas, deciduous-grass-scrub land cover types and lower occupancy in less tolerant states and highly populated areas because of direct competition for food and space (*Brashares, Arcese & Sam, 2001*; *Rangarajan, 2001*; *Parks & Harcourt, 2002*; *Karanth et al., 2010*). We performed pair-wise correlation tests to screen variables for multicollinearity. The occupancy approach accounts for non-detection of species during surveys and inability to survey some sites (see *Karanth et al., 2009*; *Karanth et al., 2010* for additional details). The probability of extinction was calculated as (1- probability of occurrence) (*Karanth et al., 2010*). We derived leopard extinction probabilities for three separate major landscapes (Western Ghats, central India and north India) as these regions strongly represented our genetic sampling. These extinction probabilities were compared to the genetically derived estimates.

## RESULTS

### Individual identification of leopards from north Indian landscape

Of the 778 large carnivore faecal samples collected from TAL, we identified 195 faeces to be of leopard origin (25%) using species-specific PCR assays (*Mondol et al., 2014*; *Maroju et al., 2016*). In addition, 457 samples were ascertained to be of tiger (59%) and remaining 126 faecal samples did not produce any result (16%) for either of these large felids, possibly due to poor quality DNA. We amplified 13 microsatellite loci panel on these 195 genetically confirmed leopard faecal samples, and after data validation through multiple repeats generated nine or more loci data from 65 faecal DNA. Subsequently, we identified 56 unique leopard individuals from the 65 samples, whereas nine individuals were ascertained as 'genetic recaptures'. The mean allelic dropout rate for these loci was found to be 0.05, whereas mean false allele rate for all the 13 loci was 0.04, indicating this 13 loci panel has low genotyping error rates. Amplification success ranged between 41–100% from

leopard faecal DNA. None of the loci were found to deviate from the Hardy-Weinberg equilibrium and there were no evidence for strong linkage disequilibrium between any pair of loci. Cumulative $PID_{sibs}$ and $PID_{unbiased}$ values were found to be $3.91*10^{-6}$ and $2.73*10^{-16}$, respectively, indicating a strong statistical support for unambiguous individual identification. Summary statistics for these samples collected across Terai-Arc landscape is provided in Table 1. We identified 26, 21 and nine unique leopard individuals from the states of Uttarakhand, Uttar Pradesh and Bihar, respectively. As the data generated from north India is comparable to the earlier data, we added this 56 unique leopard data to 143 individual genotypes described in Mondol et al. (2014), and overall 199 unique unrelated leopards were used in subsequent population structure, genetic variation and demography analyses.

## Leopard population structure and genetic variation across India

Our sampling strategy targeted countrywide leopard populations to assess population structure and genetic variation. From 199 final unique leopard genotypes we removed four samples representing the eastern and northeast India ($n = 2$ from the states of West Bengal and Assam each, respectively) from further analyses as they represented inadequate sampling from these regions. Bayesian clustering analysis using 13 microsatellite data from the remaining 195 wild leopard individuals showed four distinct genetic subpopulations ($K = 4$, see Fig. S1), as presented in Fig. 1. The DAPC analyses identified five different clusters ($K = 5$) using the Bayesian Information Criterion (Fig. S2). However, out of these genetic clusters two of them were overlapping with each other. Overall, both analyses showed the same pattern of four genetic subpopulations. Majority of the samples showed respective group-specific ancestry, with Western Ghats samples representing the first group (henceforth WG, $n = 65$), the Deccan Plateau-Semi Arid region forming the second (henceforth DP-SA, $n = 66$), the samples from Shivalik region covering parts of Himalaya and western parts of upper Gangetic plains making the third group (henceforth SR, $n = 38$), and finally samples from the Terai region covering eastern part of upper and western part of the lower Gangetic Plains samples forming the fourth one (henceforth TR, $n = 26$), respectively (Fig. 1). However, small number of samples ($n = 18$) distributed among the four subpopulations showed mixed ancestry. Subsequent analyses revealed that these leopard subpopulations are genetically differentiated ($F_{st}$ and $G_{st}$) at low, but significant levels (Table 2) for all four populations. The $F_{st}$ value among these populations ranged between 0.028–0.115, whereas the $G_{st}$ value between 0.023–0.104 (Table 2).

Analyses and with 13 microsatellite loci among the four genetic subpopulations showed a higher mean number of alleles ($NA_{WG} = 11.77$ (S.D. 3.85), $NA_{DP-SA} = 10.46$ (S.D. 2.71)) and observed heterozygosity ($H_{oWG} = 0.81$ (S.D. 0.08), $H_{oDP-SA} = 0.8$ (S.D. 0.08)) in Western Ghats and Deccan Plateau-Semi Arid subpopulations, when compared with samples from Shivalik and Terai region subpopulations ($NA_{SR} = 08.46$ (S.D. 2.41), $NA_{TR} = 05.00$ (S.D. 1.84) and $H_{oSR} = 0.40$ (S.D. 0.14), $H_{oTR} = 0.36$ (S.D. 0.28), respectively) (see Table 3 for details). However, the allelic size range values were similar in all populations (Table 3). Western Ghats and Deccan Plateau- Semi Arid subpopulations

**Table 2   Genetic differentiation (pairwise $F_{st}$ and $G_{st}$) for four leopard subpopulations in the Indian subcontinent.** The upper diagnoal presents the pairwise $G_{st}$ values whereas the lower diagnoal presents the pairwise $F_{st}$ values.

| | Western Ghats ($n = 65$) | Deccan Plateau-Semi Arid ($n = 66$) | Shivalik ($n = 38$) | Terai ($n = 26$) |
|---|---|---|---|---|
| **Western Ghats** ($n = 65$) | – | 0.023[*] | 0.039[*] | 0.091[*] |
| **Deccan Plateau-Semi Arid** ($n = 66$) | 0.028[*] | – | 0.045[*] | 0.104[*] |
| **Shivalik** ($n = 38$) | 0.048[*] | 0.05[*] | – | 0.073[*] |
| **Terai** ($n = 26$) | 0.103[*] | 0.115[*] | 0.089[*] | – |

Notes.
[*]$p$ value $= 0.001$.

showed higher number of private alleles (2.38 and 0.85, respectively) when compared to Shivalik and Terai subpopulations (0.46 and 0.15, respectively) (Table 3).

## Detection of demographic change

We used microsatellite data to investigate signals of demographic changes in each of the four leopard genetic subpopulations across the subcontinent. Both qualitative approaches, the EWCL and the M-ratio methods indicate signatures of population bottleneck. The EWCL approach implemented in the program BOTTLENECK shows 8-10 loci with heterozygote excess depending on the mutation models used, suggesting a loss of rare alleles through population decline for all four subpopulations. Similarly, the M-ratio approach also shows a low ratio between number of alleles ($N_A$) and the allelic size range in all four subpopulations (M-ratio$_{WG}$-0.37 (S.D. 0.09); M-ratio$_{DP-SA}$-0.38 (S.D. 0.09); M-ratio$_{SR}$-0.33 (S.D. 0.09); M-ratio$_{TR}$-0.29 (S.D. 0.15)), indicating signatures of population bottleneck.

In the quantitative MSVAR approach, models with exponential decline scenarios show consistently that the posterior distributions for log (N0) is always lower than log (N1) for all four subpopulations, indicating population decline for leopards across the subcontinent (Table 4 and Fig. 2). Further quantification revealed that the current effective size is varyingly low (12–25%) than the historical effective size, with Western Ghats, Deccan Plateau-Semi Arid, Shivalik and Terai regions losing approximately 75%, 90%, 90% and 88% of their leopard population, respectively (Table 4 and Fig. 2).

Our subsequent analyses also revealed distributions that suggested recent time of declines in all four populations of leopards (Table 4, Fig. 2). The north Indian subpopulations (Shivalik and Terai) and the Deccan Plateau-Semi Arid population showed the most recent decline occurred about 120–125 years before present, respectively. However, the Western Ghats population indicated potential decline around 200 years ago (Table 4 and Fig. 2).

## Leopard occurrence and distribution

We examined the factors influencing leopard distribution at a countrywide scale, where the top ranked model incorporating 28 covariates suggested a wide distribution of habitat types (described in *Karanth et al., 2009*; *Karanth et al., 2010*). The model also indicated a positive influence of protected areas, higher cultural tolerance of people and negative influence of higher human population densities and (details in *Karanth et al., 2009*).
**Table 3  Subpopulation-wise summary statistics (based on 13 microsatellite loci) for Indian leopards.**

| Locus | Western Ghats (n = 65) | | | | | Deccan Plateau-Semi Arid (n = 66) | | | | | Shivalik (n = 38) | | | | | Terai (n = 26) | | | | |
|---|---|---|---|---|---|---|---|---|---|---|---|---|---|---|---|---|---|---|---|---|
| | $N_A$ | ASR | $H_E$ | $H_O$ | PA | $N_A$ | ASR | $H_E$ | $H_O$ | PA | $N_A$ | ASR | $H_E$ | $H_O$ | PA | $N_A$ | ASR | $H_E$ | $H_O$ | PA |
| FCA230 | 13 | 36 | 0.88 | 0.86 | 03 | 10 | 22 | 0.78 | 0.80 | 01 | 08 | 24 | 0.83 | 0.23 | 00 | 05 | 26 | 0.52 | 0.65 | 00 |
| FCA309 | 17 | 42 | 0.90 | 0.87 | 05 | 11 | 30 | 0.78 | 0.81 | 00 | 08 | 18 | 0.81 | 0.46 | 00 | 06 | 10 | 0.82 | 0.32 | 00 |
| FCA232 | 13 | 36 | 0.85 | 0.84 | 03 | 09 | 18 | 0.68 | 0.72 | 00 | 09 | 26 | 0.78 | 0.42 | 01 | 07 | 26 | 0.78 | 0.46 | 00 |
| FCA090 | 14 | 30 | 0.85 | 0.84 | 02 | 08 | 18 | 0.78 | 0.87 | 00 | 09 | 30 | 0.86 | 0.36 | 02 | 02 | 10 | 0.47 | 0.00 | 00 |
| FCA052 | 12 | 32 | 0.84 | 0.89 | 02 | 11 | 22 | 0.82 | 0.84 | 01 | 08 | 20 | 0.83 | 0.48 | 00 | 06 | 22 | 0.84 | 0.43 | 00 |
| FCA672 | 19 | 40 | 0.90 | 0.89 | 09 | 10 | 26 | 0.82 | 0.75 | 01 | 06 | 16 | 0.65 | 0.50 | 00 | 07 | 20 | 0.64 | 0.77 | 00 |
| FCA279 | 11 | 26 | 0.81 | 0.75 | 01 | 14 | 26 | 0.87 | 0.81 | 00 | 15 | 28 | 0.90 | 0.67 | 01 | 06 | 18 | 0.78 | 0.92 | 00 |
| FCA126 | 14 | 26 | 0.87 | 0.83 | 03 | 13 | 30 | 0.88 | 0.89 | 00 | 09 | 22 | 0.76 | 0.18 | 00 | 04 | 12 | 0.11 | 0.74 | 00 |
| msFCA391 | 07 | 28 | 0.83 | 0.67 | 00 | 08 | 32 | 0.81 | 0.81 | 01 | 07 | 24 | 0.78 | 0.57 | 00 | 07 | 32 | 0.75 | 0.13 | 01 |
| msHDZ170 | 09 | 20 | 0.84 | 0.79 | 00 | 10 | 22 | 0.75 | 0.88 | 01 | 10 | 36 | 0.76 | 0.16 | 02 | 02 | 02 | 0.29 | 0.0 | 00 |
| msFCA441 | 07 | 36 | 0.75 | 0.79 | 02 | 08 | 28 | 0.65 | 0.55 | 00 | 08 | 40 | 0.82 | 0.38 | 00 | 05 | 28 | 0.66 | 0.40 | 01 |
| msFCA506 | 12 | 32 | 0.86 | 0.90 | 01 | 17 | 56 | 0.83 | 0.82 | 04 | 09 | 24 | 0.83 | 0.33 | 00 | 06 | 22 | 0.79 | 0.40 | 00 |
| msFCA453 | 05 | 20 | 0.63 | 0.67 | 00 | 07 | 32 | 0.69 | 0.80 | 02 | 04 | 20 | 0.65 | 0.36 | 00 | 02 | 16 | 0.43 | 0.14 | 00 |
| Mean (SD) | 11.77 (3.85) | 31.08 (6.69) | 0.83 (0.07) | 0.81 (0.08) | 2.38 | 10.46 (2.71) | 27.85 (9.36) | 0.78 (0.07) | 0.80 (0.08) | 0.85 | 08.46 (2.41) | 25.23 (6.64) | 0.79 (0.07) | 0.40 (0.14) | 0.46 | 05.00 (1.84) | 18.77 (8.21) | 0.65 (0.17) | 0.36 (0.28) | 0.15 |

**Notes.**

$N_A$, No. of alleles;  ASR,  Allelic size range; $H_E$, Expected heterozygosity; $H_O$, Observed heterozygosity;  PA,  Private alleles.

**Table 4  Comparison of different demographic decline analyses results for different subpopulations of leopards across India.**

| Method | Analysis type | Demographic signal | | | | |
|---|---|---|---|---|---|---|
| | | Model | Western ghats | Deccan plateau-Semi arid | Shivalik | Terai |
| Bottleneck | | IAM | Heterogygosity excess for 13 loci | Heterogygosity excess for 10 loci | Heterogygosity excess for 12 loci | Heterogygosity excess for 11 loci |
| | | SMM | Heterogygosity excess for 01 loci | Heterogygosity excess for 02 loci | Heterogygosity excess for 06 loci | Heterogygosity excess for 08 loci |
| | Qualitative | TPM | Heterogygosity excess for 07 loci | Heterogygosity excess for 07 loci | Heterogygosity excess for 09 loci | Heterogygosity excess for 10 loci |
| M ratio | | | 0.37 (SD 0.09) | 0.38 (SD 0.09) | 0.33 (SD 0.09) | 0.29 (SD 0.15) |
| Storz-Beaumont method | Quantitative | | Decline—75% Time—~200 years ago | Decline- 90% Time—~125 years ago | Decline- 90% Time—~125 years ago | Decline—88% Time—~120 years ago |
| Extinction probability | Quantitative | Occupancy | 0.17 | 0.21 | 0.37 | |

Areas with cultivated land, barren areas, deciduous forests and rural–urban were strongly associated with higher leopard occurrence. Naive estimated occupancy was 0.52, whereas model estimated probability of occupancy was significantly higher at 0.68, suggesting that leopards are still widely distributed (Fig. 3) in India compared to most other large mammals (as suggested in *Karanth et al., 2010*). When compared among the overall three major sub-regions (north India (NI), Deccan Plateau-Semi Arid and Western Ghats), we find that average estimated occurrence was lowest in the north India ($Psi_{NI} = 0.63 \pm 0.01$, Range: 0.05–1.00, 384 cells) compared to Western Ghats ($Psi_{WG} = 0.83 \pm 0.02$, Range: 0.23–1.00, 90 cells) and Deccan Plateau-Semi Arid ($Psi_{DP-SA} = 0.79 \pm 0.005$, Range: 0.25–1.00, 818 cells). Overall, average estimated Psi was $0.74 \pm 0.006$ (1,292 cells).

# DISCUSSION

To the best of our knowledge, this is probably the first and most exhaustive study on leopard population genetics and demographic patterns in the Indian subcontinent. Except the eastern and northeast Indian landscape, where our sampling intensity was less all other regions are well covered in this study. Our genetic analyses with microsatellite data collected across the subcontinent reveal four genetic subpopulations of leopards in India: the Western Ghats, Deccan Plateau-Semi Arid landscape, hill region of north India (Shivalik) and Terai or flat region of north India. While there was some amount of mixed genetic signal across different genetic subpopulations, they were clearly separated as different groups (Fig. 1). These genetic groups mostly correspond to respective biogeographic zones of India, with Western Ghats and combination of Deccan Plateau-Semi Arid forms two subpopulations, whereas the north Indian subpopulation of Shivalik and Terai are parts of the Himalayan and Gangetic Plains zones, respectively.

Based on these patterns, we presume that these genetic clusters were formed due to restricted gene flow along major habitat type differences between these biogeographic

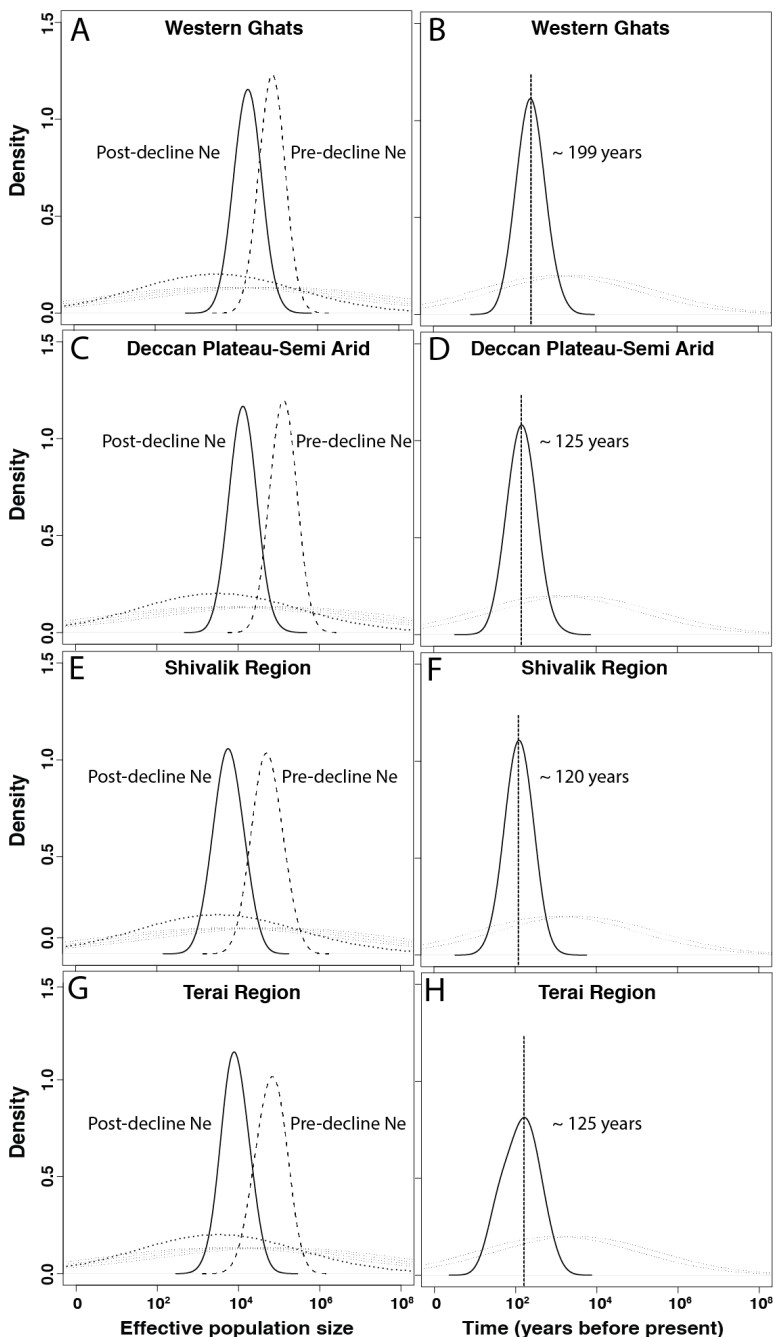

**Figure 2** **Demographic history of Indian leopards (*Panthera pardus fusca*).** A, C, E and G show the posterior distributions for leopard population size changes for different subpopulations, based on 13 microsatellite loci using Storz and Beaumont approach. The dashed and solid lines represent posterior distributions of ancestral and present effective population sizes. The priors are represented by the dotted line. B, D, F, and H represent the posterior distribution for the time since the leopard population decline started for corresponding subpopulations. The priors are shown by the dotted lines.

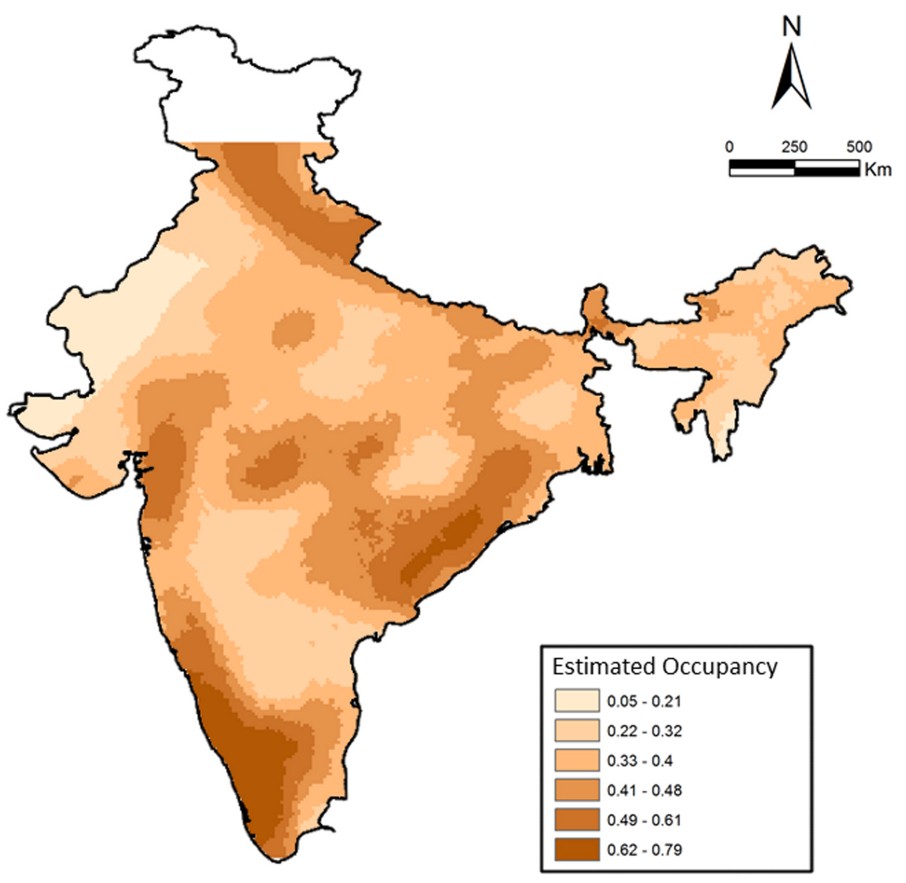

**Figure 3** **Patterns of leopard occurrence in India based on the analysis of questionnaire surveys.** The map shows a gradient of estimated cell-wise occupancy probabilities created through spatial kriging.

zones. For example, the difference between habitat types of large contiguously forested Western Ghats (largely moist deciduous habitat) and the Deccan Plateau-Semi Arid (mostly dry deciduous habitat) probably lead to genetic differences between leopards from these regions. Similarly, difference in habitat types in 'Bhabar' habitats of Shivalik (hilly rugged terrains, large trees, high leopard abundance) and Terai region (flat terrain, grassland, low leopard abundance) (*Johnsingh et al., 2004*), has possibly led to reduced gene flow between these two areas. Such habitat-driven reasons meaningfully explain the genetic differentiation between leopards from these landscapes. These four subpopulations were genetically differentiated by low, but significant levels (Table 2). Previous studies on tigers (*Mondol, Karanth & Ramakrishnan, 2009*; *Mondol, Bruford & Ramakrishnan, 2013* subcontinent scale) as well as leopards (*Dutta et al., 2013*, central Indian landscape) suggested long-distance movement as a potential cause for low genetic differentiation between populations. Leopards are known to disperse long distances (*Ropiquet et al., 2015*; *Farhadinia et al., 2018*) and human-leopard conflict driven translocation is common in many parts in India (*Athreya et al., 2010*; *Odden et al., 2014*). Together, natural dispersal abilities and 'human mediated gene flow' through translocations might be responsible for

the low genetic differentiation among leopard subpopulations across the subcontinent. Earlier work in central Indian landscape (*Dutta et al., 2013*) suggested a reduction in gene flow at recent times due to habitat destruction, but our study did not focus to answer such questions. Future studies should focus on using historical samples (museum skins, bones etc.) to assess any possible change in gene flow among leopard populations (For example see *Martinez-Cruz, Godoy & Negro, 2007*; *Valdiosera et al., 2008*; *Lorenzen et al., 2011*; *Mondol, Bruford & Ramakrishnan, 2013*) at subpopulation levels across the country.

However, our demography analyses with genetic data indicate strong decline in leopard population size across all four genetic subpopulations. Results with both qualitative (bottleneck and M-ratio approach) as well as quantitative (Storz and Beaumont approach) analyses revealed strong, but varying signals of demographic decline in all four subpopulations (Table 4). The Deccan Plateau-Semi Arid, Shivalik and Terai subpopulations show 90%, 90% and 88% decline in population size, respectively, whereas the Western Ghats subpopulation show relatively less (75%) decline in population size (Table 4). Leopards are vulnerable to conflict and poaching due to their close associations with human habitations (*Gavashelishvili & Lukarevskiy, 2008*; *Athreya et al., 2010*; *Balme, Slotow & Hunter, 2009*). The Western Ghats retains possibly the largest contiguous forested landscape with multiple interconnected protected areas, whereas the other regions have lot of human activities, possibly affecting leopard populations living in them. Further, the ecological data based occupancy analysis showed extinction probabilities of 0.37, 0.21 and 0.17 for north India, Deccan Plateau-Semi Arid and Western Ghats landscape, respectively (Table 4). It is possible that this discrepancy in the magnitudes of decline based on genetic and ecological models is because the ecological methods are more spatial. The inferences from this model are dependent on temporal differences in leopard occupancy. However, if densities of leopards were high in the past, loss of even small habitats could result in the loss of many individuals. Since no quantitative comparisons for leopard density between the Western Ghats, Deccan Plateau-Semi Arid and north India is currently available, we cannot conclusively infer the former, but further research should investigate leopard densities and their temporal changes across the country. Finally, this decline pattern also roughly corroborates with 83–87% leopard range loss in Asia (*Jacobson et al., 2016*), indicating that habitat loss is a contributing factor towards the population decline.

The magnitude of decline for leopards found in this study is contrasting to some of the earlier leopard studies in the subcontinent (for example ecological work by *Harihar, Pandav & Goyal, 2011*, and genetic work by *Dutta et al., 2013*) and eastern Africa (*Spong, Johansson & Bjorklund, 2000* in Tanzania), which suggest stable or increasing local leopard population trends. This is certainly possible as many of these studies were conducted inside protected areas, where leopard population dynamics depends on presence/absence of other large carnivores (tiger, dhole etc.). Given that only 11% of Indian leopard distribution is within protected area network (*Jacobson et al., 2016*), it is challenging to truly understand the population trends at country level. Our sampling at subcontinent scale is thus indicating the actual patterns of population demography that is difficult to assess based on ecological/genetic studies at local level.

Another important finding is the relatively recent timing of decline for all the leopard subpopulations in the subcontinent. Our results suggest median leopard decline timing between 120–200 years across four genetic subpopulations (Table 4). Except Western Ghats (decline timing of ∼200 years), all other subpopulations indicate much recent population decline (Deccan Plateau-Semi Arid ∼125 years, Shivalik ∼120 years and Terai ∼125 years). When compared with other sympatric, endangered species in the subcontinent (for example tiger decline ∼200 years ago; *Mondol, Karanth & Ramakrishnan, 2009*) or Asian region (for example Orangutan- ∼210 years, *Goossens et al., 2006*; Giant panda- ∼250 years, *Zhu et al., 2010*) this still seems to be much recent event. Other wide-ranging carnivores across the globe (for example European wolves *Aspi et al., 2006*; African wild dog- *Marsden et al., 2012*; Eurasian badgers-*Frantz et al., 2014* etc.) too faced much longer decline period than leopards. One plausible explanation could be recent increases in leopard-human conflict (*Athreya et al., 2010*; *Karanth & Kudalkar, 2017*) and poaching intensity due to large demand of leopard body parts in the illegal wildlife markets (*Raza et al., 2012*; *WPSI, 2017*). Historically, major leopard hunting events had been recorded across the Indian subcontinent during Mughal times (about 500–600 years ago), followed by colonial British bounty-hunting rule between 1850–1920 (*Rangarajan, 2005*). However, large-scale landscape modification and fragmentation by humans during the last century (central India-*Rangarajan, 1999*, north India-*Rangarajan, 2005*), coupled with poaching and conflict may have resulted in much recent loss of leopard populations across the country. We lack comprehensive data, both at historical as well as modern scales to investigate the true causes behind such patterns of differential population decline timing. For example, *Dutta et al. (2013)* showed that during last three centuries severe changes in landscape characteristics (Settlement, villages, wild lands, human density) have occurred in the central Indian leopard habitats. However, we lack information on hunting and conflict levels during this time from these regions. Future efforts should generate this important information to get an idea of the scenarios leading to such strong decline in a wide-ranging species like leopard. Finally, it is important to point out that in this study we have only explored relatively simple decline scenarios during demographic modelling. Future studies should evaluate more detailed, computationally intensive demographic analyses with genome wide molecular markers (For example, see *Frantz et al., 2014*; *Nater et al., 2015*) for better understanding of complex decline scenarios.

Finally, another important aspect of the results from this study is that despite severe decline (Table 4) and small, but significant population structure (Fig. 1B, Table 2) leopards still retain high genetic variation in the Indian subcontinent. We found that leopard genetic variation across four genetic subpopulations is similar and comparable to eastern and southern Africa (*Spong, Johansson & Bjorklund, 2000*; *Uphyrkina et al., 2001*; *McManus et al., 2015*), and higher than Arabian (*Ilani, 1981*; *Perez, Geffen & Mokady, 2006*) and Amur leopards (*Uphyrkina et al., 2001*; *Sugimoto et al., 2014*). The higher levels of variation could possibly be attributed to still relatively large population size, high pre-bottleneck genetic variation and potential historical gene flow across large landscapes.

## CONCLUSION

While leopards are relatively easier to study than other sympatric carnivores like tigers due to their ubiquitous presence, studies on their population size, trend and dynamics are limited, particularly in outside protected areas. In fact, due to their broad geographic distribution, leopard populations are perceived to be stable, with current IUCN Red List status of 'vulnerable'. However, both historical records and recent conflict with humans suggest potentially declining population trends. Using genetic data, we reveal a strong signal of population decline (between 75–90%) across different habitats in the Indian subcontinent over the last 120–200 years. We demonstrate population decline in a wide-ranging and, commonly perceived as locally abundant species like the leopard, suggesting that leopards demand similar conservation attention like tigers in India. While we are unable to corroborate these population decline patterns with leopard census data, our results suggest that it will be important to generate such ecological abundance estimates for leopard populations in the near future. This work also emphasizes the importance of similar work on wide-ranging species, as it is possible that other species like the leopard may show population decline, especially in the context of the Anthropocene.

## ACKNOWLEDGEMENTS

We acknowledge the Director, Dean and Nodal Officer of Wildlife Forensics and Conservation Genetics Cell of Wildlife Institute of India for their support in this work. We thank Dr. Uma Ramakrishnan of National Centre for Biological Sciences for providing reference leopard samples. Mr A. Madhanraj has provided critical support in genotyping facility and Mr. Debanjan Sarkar helped with GIS in the laboratory. We thank all the lab members of Wildlife Forensic and Conservation Genetics cell and especially Meercat lab for productive discussions and valuable comments. We also thank Dr. S.K. Gupta and Dr. S.P. Goyal for logistic support; our field assistants Annu, Bura, Abbhi, Ranjhu and Imam for their effort in the field. Finally, we thank two reviewers and the editor for their critical suggestions to improve our earlier version of the manuscript.

### Funding

This research was funded by Wildlife Conservation Trust-Panthera Global Cat Alliance Grants and Department of Science and Technology, Government of India grant no EMR/2014/000982. Samrat Mondol was supported by the Department of Science and Technology INSPIRE Faculty Award (No.IFA12-LSBM-47) and Krithi Karanth was supported by Centre for Wildlife Studies, Wildlife Conservation Society-New York and Oracle. The funders had no role in study design, data collection and analysis, decision to publish, or preparation of the manuscript.

### Grant Disclosures

The following grant information was disclosed by the authors:

Wildlife Conservation Trust-Panthera Global Cat Alliance Grants and Department of Science and Technology, Government of India: EMR/2014/000982.
Department of Science and Technology INSPIRE Faculty Award: IFA12-LSBM-47.
Centre for Wildlife Studies, Wildlife Conservation Society-New York and Oracle.

## Competing Interests

The authors declare there are no competing interests.

## Author Contributions

- Supriya Bhatt and Suvankar Biswas performed the experiments, analyzed the data, prepared figures and/or tables, and approved the final draft.
- Krithi Karanth analyzed the data, prepared figures and/or tables, and approved the final draft.
- Bivash Pandav conceived and designed the experiments, authored or reviewed drafts of the paper, fieldwork supervision, and approved the final draft.
- Samrat Mondol conceived and designed the experiments, analyzed the data, prepared figures and/or tables, authored or reviewed drafts of the paper, made earlier leopard data available, and approved the final draft.

## Animal Ethics

The following information was supplied relating to ethical approvals (i.e., approving body and any reference numbers):

The work conducted here is using non-invasive samples (faeces) collected from the wild with appropriate permissions. Due to the non-invasive nature of the work, no ethical approval is required.

## Field Study Permissions

The following information was supplied relating to field study approvals (i.e., approving body and any reference numbers):

Forest Departments from the states of Uttarakhand, Uttar Pradesh and Bihar provided required permission for field sampling (90/5-6, Uttarakhand Forest Department) (1127/23-2-12(G) and 1891/23-2-12(G), Uttar Pradesh Forest Department) (Wildlife-589, Bihar Forest Department).

## Data Availability

The raw data used in this study are available in the

Supplemental Files and at Dryad: Bhatt, Supriya et al. (2020), Genetic analyses reveal population structure and recent decline in leopards (*Panthera pardus fusca*) across Indian subcontinent, v2, Dryad, Dataset, 10.5061/dryad.v6wwpzgrg.

## Supplemental Information

Supplemental information for this article can be found online at http://dx.doi.org/10.7717/peerj.8482#supplemental-information.

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
