# Peer review of "Genetic analyses reveal population structure and recent decline in leopards (Panthera pardus fusca) across the Indian subcontinent"

_PeerJ, doi:10.7717/peerj.8482_

## Round 0.1 · original submission · Minor Revisions

I agree with the reviewers that this is a nicely written and timely manuscript, and with some suggested revisions it stands to be even better. Please address the suggestions and make the corrections and we look forward to seeing a revised version soon. Although a reviewer suggested "major revisions" his comments suggest otherwise and tend towards "minor" corrections and some thought into structure and content.

Reviewer 1 ·

Basic reporting

The authors have compiled a large dataset of leopards from 4 landscapes in India to address a number of issues on population structure and bottlenecks. The paper is nicely written, with proper structure and language. I do not see any huge problem in the paper, but provided some comments to help authors to improve their interpretation and applicability.

Also, I believe that authors must consider these four main issues:
1. The paper length is long, particularly in Discussion. Authors are strongly recommended to remove redundant or general parts in Discussion, as they ruin their scope of paper and its exciting findings.
2. Also, I feel that the authors should be a bot careful in interpretation of their findings in conservation. Although the leopard population and range have been reduced, but the heterozygosity of Indian leopards is still high, perhaps highest among Asian leopards, and the extinction rate was calculated as up to 20% for two Indian population, which is far below than many other leopard populations. Thus, in the context of leopard conservation, I think a more careful interpretation is needed.
3. Also, authors are recommended to provide a proper background on genetic studies of leopards, at least in Asia, to provide a comparative baseline for their findings.
4. My final major issue is that more work needed to interpret population structure and combine them with proper ecological data from other literature to rationalize one of the main findings of this paper, which is pop structuring.

Experimental design

Lab experiments have been done properly, field sampling too.
I was curious to know what specific markers were used for species identification (rather than referring to Mondol's paper).
The paper objectives were clear, while I preferred to see all those 4 objectives as some sort of research hypotheses, as what did they expect a priori to find during this survey? Did they expect such a population structure before the analyses, and why?

Validity of the findings

I found findings robust, although some additional tests have been proposed (Ne estimator or PCA) to support their claims.

Additional comments

Abstract: Nicely written, but cannot we consider grey wolf more widespread and highly adaptable than leopards?

Introduction
The introduction is nice and easy to read, but I feel there are redundant sections, particularly for the first two paragraphs with too many citations, many of them really not needed. Although the paper is about the genetic, but the authors have not provided a proper genetic background of leopards, at least within the Asian context, so the reader could learn what we know about leopard genetic and what are the genetic gaps? Based on this background, in following paragraphs, you can narrow your debates down to India.
I recommend you to use these papers for a thorough review of genetic status of leopards in Asia:

Asad, M., Martoni, F., Ross, J.G., Waseem, M. and Paterson, A.M., 2019. Assessing subspecies status of leopards (Panthera pardus) of northern Pakistan using mitochondrial DNA. PeerJ, 7, p.e7243.

Farhadinia, M.S., Farahmand, H., Gavashelishvili, A., Kaboli, M., Karami, M., Khalili, B., Montazamy, S., 2015. Molecular and craniological analysis of leopard, Panthera pardus (Carnivora: Felidae) in Iran: Support for a monophyletic clade in Western Asia. Biol. J. Linn. Soc. 114, 721–736.

Uphyrkina, O., Johnson, W.E., Quigley, H.B., Miquelle, D.G., Marker, L., Bush, M.E., O’Brien, S.J., 2001. Phylogenetics, genome diversity and origin of modern leopard, Panthera pardus. Mol. Ecol. 10, 2617–2633.

Paijmans, J.L., Barlow, A., Förster, D.W., Henneberger, K., Meyer, M., Nickel, B., Nagel, D., Havmøller, R.W., Baryshnikov, G.F., Joger, U. and Rosendahl, W., 2018. Historical biogeography of the leopard (Panthera pardus) and its extinct Eurasian populations. BMC evolutionary biology, 18(1), p.156.

First para: Do you really need all these citations for these generally agreed sentences? I suggest removing non-necessary ones.
Second para: Many of these arguments are not really within the scope of your paper, talking about IUCN status of Amur leopard is not relevant to your work in India. Please consider proper briefing of the whole para.
L110: But Dutta’s work is not at local scale, it is central India, which could be interpreted as regional scale, based on your work.

Methods

Major issues:
Did you use methods of rarefaction that accommodates the effect of small sample size for calculating unbiased allelic richness, i.e. the number of alleles in a sample using HP-RARE 1.0 (Kalinowski, 2005)?
What is Ne of the current population? I suggest you to use the molecular co-ancestry method of Nomura (2008), as implemented in NeEstimator V2 (Do et al., 2014) using the linkage disequilibrium method.
Based on Ne, you then can find published ratios between N/Ne to project the current population size of leopards in India, and compare it to Jhala’s 2015 report of 7000 something. I can suggest you two ratios of 0.11 based on different taxa (Frankham, 1995) and 0.42 calculated from felid case studies (Spong et al., 2000) between census and effective population sizes (N/Ne).
As you believe there are 4 populations, it implies a separation in breeding. I suggest you to calculate the number of migrants successfully entering a population per generation (Nm) as well, as a proof for your claim for populations.
L215: For genetic structure, you have only used STRUCTURE.
I suggest you adding the principal component analysis (PCA) using ‘adegenet’ package (Jombart and Ahmed, 2011) in R (R Development Core Team, 2013).

Also, a main topic in the paper is the reasons behind these population structuring, which I think you need to do additional tests for different hypotheses. For example, another important metircs for population structure is the Mantel test for Matrix Correspondence in GENALEX 6.5, to evaluate isolation by distance. It can tell you more about if isolation between populations is because of distance or because of human barriers (i.e. distance are not large, but human barriers are prominent).

L282: Can you clarify the reason for choosing this grid size?
L302: Citation needed for this method
In occupancy modelling, proper details needed for model building, selection and validation procedures (AIC/over-dispersion etc).

Results
L311 It is a bit strange why no other animal, particularly canids were found in your large dataset, any justification? I expected to see lots of dogs though
L346: A report of private alleles for each population would be great here.
L363-364: It is methods, not results.
L389-394: I suggest reporting psi as below: psi=XX ± SE XX. More details should go to a table, not needed here.
Also, what happened to detection probability and what covariates are associated with that?

Discussion:
It is long, some paragraphs are up-to a page long. Please consider make it shorter, and split long sentences and para into smaller thematic ones.
L402-405: Population, subpopulation and cluster? Please be consistent in using terms across your paper, different terms if used interchangeably will confuse readers.
L408: Mantel test for Isolation by Distance hypothesis will help you to find out better about the reason of separation.
L412: How habitat heterogeneity might cause genetic differences? Not clear. I understand that intraguild competition with tigers can affect pop density of leopards, but how he story ends up in genetic differences?
L414-416: Already mentioned in Results.
L433 and L43 and L437 population OR subpopulation? Please be consistent
L442 Is 0.17 or 0.22 a high extinction rate in current world that many species are suffering higher rates? With preceding debates, I expected higher extinction rate, but it seems that some of Indian leopard populations are doing well.
L454: Citation needed
L454: After such a long para, I should admit that I was lost somewhere half way through, very difficult to keep the track of the narration.
L458: Was T Dutta’s work across a multi PA landscape? I recall it was not within a reserve
L461: Other factors like?
L465-471 This is unjustifiable. To confirm that Indian leopards have decreased, evidence of panda and orangturan and wolves is absolutely irrelevant. Yes, we know many species are declining, but that does not mean a declining trend for another species as an evidence for something else. Please consider reworking or even removing and stick to your data and conclusions
L492: For example (not for e.g.)
L500 ditto
L505 It is not Africa, it is only Tanzania
L506 Did Ilani and Perez reported anything about genetic variability of leopards? I doubt
L506 I recommend to compare your results with another Asian leopard population in west Asia, based on this paper

Rozhnov, V. V, Lukarevskiy, V.S., Sorokin, P.A., 2011. Application of molecular genetic characteristics for reintroduction of the leopard (Panthera pardus) in the Caucasus. Dokl. Biol. Sci. 437, 280–285.

L512 Do you really mean that data on leopard ecology is limited? There are tons of papers out there, making leopards as one of the most studied large carnivores in Asia

Finally, in Discussion, to argue about separation of these 4 populations, I think you need to provide some movement data. I was wondering how long a leopard can disperse or how far are these 4 landscapes that leopards cannot reach from one to the other one?
There are three key papers that you can adopt some movement and dispersal data from:

1. This one shows how far translcoated leopards move (in one case nearly 100 km), tough translocated animals are not usually a proper representative of true dispersal:
Odden, M., Athreya, V., Rattan, S. and Linnell, J.D., 2014. Adaptable neighbours: movement patterns of GPS-collared leopards in human dominated landscapes in India. PLoS One, 9(11), p.e112044.

2. This one shows how far leopards can be translocated to be still within their genetic subpopulation?
Ropiquet, A., Knight, A.T., Born, C., Martins, Q., Balme, G.A., Kirkendall, L., Hunter, L.T.B., Senekal, C., Matthee, C.A., 2015. Implications of spatial genetic patterns for conserving African leopards. C. R. Biol. 338, 728–737.

3. This paper shows dispersal distance of a young leopard, a class in the species with highest rate of dispersal, moving through human landscapes (ca. 80 km), in an area with leopard density similar to many parts of India (4-6 individual/100 km2):
Farhadinia, M.S., Johnson, P.J., Macdonald, D.W. and Hunter, L.T., 2018. Anchoring and adjusting amidst humans: Ranging behavior of Persian leopards along the Iran-Turkmenistan borderland. PloS one, 13(5), p.e0196602.

·

Basic reporting

1. Watch for grammatical errors, e.g., Line 26: “ its’ ” should be “ its ”
2. Line 31 – include the number of fecal samples used. You indicate 56 unique individuals, 13 loci, 143 leopards so why not include fecal samples used in your work?
3. Line 63: reword this sentence. I don’t think you mean to say that absence of large carnivores leads to significant changes in trophic cascades but rather significant changes to ecosystem structure and functioning may affect trophic dynamics, which can have cascading effects.
4. Line 74: insert “The” before “leopard”
5. Line 105 – here and in all instances change “For e.g.,” to either “For example” or “e.g.,”
a. E.g., = example given – it is redundant to use both “For example given”
6. Line 106: insert “by”  “field observations…by…Kranath et al…”
7. Line 113 – this sentence is wordy and should be rewritten for clarity. Generally, fecal samples are non-invasive, unless you are taking a fecal swab from an immobilized animal. Also minimize the use of the same word (“genetic”) multiple times in the same sentence. Reduce wordiness – e.g., “In this paper, we used fecal samples…”
8. Lines 128-133 – how were samples collected? For example, were these field surveys intentional for fecal samples or were they opportunistically collected? Foot transects, scat detection dogs, atv/vehicle?
a. Were field surveys only conducted on established trails? Roadways? If on/surrounding established pathways how far off of the road were samples collected?
b. Are these details absolutely critical? No, but will they help your reader and any other researchers interested in conducting a similar study understand how you went about obtaining your non-invasive samples? I think so.
9. Line 142: Remove excess words. Remove “have also” and “leopard”. You’ve established your target species already (leopard) and where possible make your statements active rather than passive.
10. Line 147: “used the remaining”  Insert “the”
11. Line 151-152: Reword the last sentence in this paragraph. As written this sentence is illegible.
12. Line 182: Remove “then” before alleles
13. Line 186: Replace “was” with “were” as data is the plural of datum
14. Line 187: Remove “the”
15. Line 188: Suggest to replace “good quality” with “high quality”
16. Lines 204-206: I’m confused, this sounds like it should be two separate sentences
17. Line 213: “we used the program GENECLASS 2.0…” Insert “the”
18. Line 283: double check the place of commas to separate your divisions. Should this be 3,463,322 km2?
19. Lines 306-311: You mention 126 fecal samples produced no result – does this mean no result for felid? No result for carnivore? If the latter it is misleading to state you collected 778 large carnivore fecal samples when only 652 were identified as large carnivore. Avoid unintentionally misleading your readers by stating you collected 778 fecal samples and then X% were leopard, Y% were tiger, etc
20. Line 329: Replace “country-wise” with “countrywide” or “country-wide” but be consistent with the term you use – reference Line 379.
21. Line 354: “We used microsatellite data” remove “the”
22. Line 379: “choose either “countrywide” or “country-wide” but be consistent throughout manuscript
23. Lines 407-411: Can you provide more details on “when” genetic clustering or diversification might have occurred from leopards inhabiting different biogeographic zones? For example, even a rough estimate of whether this is likely to have occurred prior to major human influence on wildlife distribution or after would help place context.
24. Line 409: For example, an earlier study…” insert “an”
25. Lines 446-449: This sentence is a bit wordy and awkward. Suggest some mild rewording, and if necessary break into multiple sentences.
26. Line 455: change to “…loss is a contributing factor of population decline.”
27. Lines 462-464: Another awkward sentence, having trouble following, particularly “…sampling at subcontinent scale is thus probably indicating the decline patterns at much larger scale.” Suggest rewording.
28. Lines 447, 478, 480, 500: see comment #5 regarding “For e.g.,”
29. 482 change “increase” to “increases”
30. Line 498: change “has possibly” to “may have”
31. Lines 490-491: trim excess words  Change “Apart from sporadic information, we…” to “We”
32. Line 492: Avoid starting sentence with abbreviations, suggest a change to “For example,…”
33. Lines 497-501: Long sentence, consider breaking this up to make it easier for the reader to digest.
34. Line 506: Be very cautious of the data presented in Uphyrkina et al. 2001 for African leopards. Samples were heavily biased towards Southern Africa and not representative of the continent. See Anco et al. 2017 – African leopards were revisited with mtDNA and novel sampling but did not include microsatellite data.
35. Line 517: Remove “Our results are interesting because”. Start with “We”.
36. Figure 1: Increase the size (legibility ) of your latitude and longitude markers.
a. Decrease the size of your symbol for “Individual leopards” to be approximately the same size as is used in the figure to donate sampling locations.
37. Figure 3: Is there an explanation why the estimated occupancy does not extend to fill the northern extend of the country shapefile you are using?

Experimental design

1. The methods you’ve mentioned and describe provide the essential information needed to replicate the study. The only major suggestion I have to increase transparency of your sampling efforts is to detail how/where samples were collected. Refer to comment #8 above. Were fecal samples the intention of field surveys or were they a byproduct of other surveys? Were they collected primarily along/near roads? If so what was the buffer distance? This information would be helpful in the design of future studies. Let’s say for example, most fecal samples were collected opportunistically on/near secondary roads. Having this information and seeing the number of fecal samples your team was able to acquire would be rather useful information in designing a study specifically to search for fecal samples.
2. The only other comment I have regarding your methods is why bother with microsatellite analyses? They are still informative, yes, but are increasingly being replaced with newer techniques that require less DNA – e.g., RADSeq, Next Gen Sequencing, etc…

Validity of the findings

1. You provide a thoughtful interpretation of your results and bring in relevant data and literature to support your speculations. I appreciate you mentioning where your limitations are and to what extend your data support your conclusions without overreaching.
2. This is an impressive dataset in both samples and geographic coverage and no doubt required the concerted efforts a large and coordinated team. Non-invasive sampling will become increasingly important as populations of threatened species, like the leopard, continue to decline and become more difficult to study using more ‘traditional’ or invasive methods.
3. If there is one major criticism I have of the paper it is the use of microsatellites. You mention this in Lines 499-501, but I would like to see these conclusions revisited with more novel analyses (RADSeq / Next Gen) as microsatellite analyses are quickly being shelved along with Sanger Sequencing techniques. The information we’re able to gather from these newer techniques can provides exceptional detail when it comes to making inferences about biogeography, anthropogenic impacts, and genetic exchange among populations. We are just beginning to scratch the surface with what we’re capable of learning through genetic techniques and I look forward to what you’ll be able to do with this sample base in future studies.

Additional comments

You have analyzed an impressive number of samples and synthesized that information into a well-put-together manuscript. With some tweaking you should have a solid paper on your hands. A bit confused why Figures 1 and 3 do not share the same spatial extent but that should be easily fixable. Good luck with your revisions!

---

## Round 0.2 · accepted · Accept

Thank you for addressing the reviewers comments and suggestions. The corrected manuscript is now ready for publication.